# Investigating External and Internal Loads in Male Older Adult Basketball Players during Official Games

**DOI:** 10.3390/jfmk7040111

**Published:** 2022-12-07

**Authors:** Daniele Conte, Federico Palumbo, Flavia Guidotti, Kestutis Matulaitis, Laura Capranica, Antonio Tessitore

**Affiliations:** 1Institute of Sport Science and Innovations, Lithuanian Sports University, 44221 Kaunas, Lithuania; 2Department of Movement, Human and Health Sciences, University of Rome “Foro Italico”, 00135 Rome, Italy; 3Department of Coaching Science, Lithuanian Sports University, 44221 Kaunas, Lithuania

**Keywords:** performance analysis, older adults, health, RPE, basketball game, IMU

## Abstract

This study aimed at assessing the external [Player Load (PL), acceleration (ACC), changes of direction (COD), JUMP, and their relative values (PL/min; ACC/min; COD/min and JUMP/min)] and internal [percentage of the peak heart rate (%HR_peak_) and the training load calculated with the session rating of perceived exertion (sRPE) method (sRPE-load)] loads of masters (senior citizen) basketball players during official games. Thirteen male basketball masters players (age: 66.6 ± 2.1 years; body mass: 89.9 ± 8.7 kg; stature: 183.7 ± 4.6 cm) were monitored during an official Lietuvos Krepsinio Veteranu Lyga (LKVL) 65-year game. Beside descriptive analysis, a chi-square goodness of fit test was adopted to assess the differences in the distribution within JUMP, ACC and COD classes of intensities (i.e., low, medium and high). The results revealed PL = 269.9 ± 83.3 AU and PL/min = 6.54 ± 1.29 AU/min. Moreover, significant differences (*p* < 0.001) in the distribution of the intensity classes were found for JUMP, ACC, and COD, with the lowest intensities as the most frequent. Finally, HR_peak_ = 81.7 ± 8.1% and sRPE-load = 148.9 ± 69.7 AU were found, with sRPE = ~3 AU. In conclusion, a low external load during an official basketball game was found compared to other basketball populations. Moreover, a high objective internal load did not correspond to a low perceived demand, which might increase the training adherence and motivation during long-term studies.

## 1. Introduction

The number of older people is increasing in European and high-income countries, with increasing life expectancy determining a demographic aging with profound implications impacting health and social care systems [1]. Aging is a complex process involving many factors (e.g., genetics, lifestyle factors, chronic diseases) that interact with one another, greatly influencing the way individuals age [2]. Globally recognized as providing health, social, environmental, and economic benefits, physical activity in advancing age has become a societal challenge [3,4].

Despite the fact that physical activity can be undertaken in a variety of ways, ranging from unstructured daily life movements and active recreation to structured physical exercise and sports, 61% of older adults unfortunately adopt inactive lifestyles [3,4,5,6]. To elicit a number of favorable responses that contribute to health and quality of life, physical activity guidelines urge older individuals to engage in multicomponent weekly physical activity encompassing at least 150 min of moderate-intensity aerobic activities, or 75 min of vigorous-intensity aerobic activities, or an equivalent combination of moderate- and vigorous-intensity activities, in addition to 10–15 repetitions of 8–10 exercises in resistance training that involve major muscle groups on two or more days [3,7]. 

Masters athletes are proposed as an ideal model for successful aging due to long-term involvement in physical activity and/or individual and/or team sports maintaining their aerobic and anaerobic functions, strength, and coordination, which have proved to be associated with life independence [8,9,10,11,12,13,14,15,16,17,18,19,20,21,22,23]. In particular, research on masters athletes focused mostly on individual sports (i.e., [8,10,12,13,14,15,21,24,25,26,27,28,29,30]), whereas limited studies specifically investigated team sports [12,31,32,33,34,35,36] and particularly masters basketball players [13,37]. However, when comparing youth, elite, and older players, it has to be taken into consideration that different physiological and technical aspects might be due to individual capabilities, as well as to the type, frequency, duration, and intensity of the training regimens of different sports [37].

Worldwide, basketball is one of the most popular team sports played at professional, competitive, and amateur levels, from youth to older ages [38,39]. Because basketball is an intermittent open-skills team sport, basketball players are required to rapidly respond to the changing environment (e.g., interactions with teammates and opponents, time pressure, referee decisions), while finely controlling their body movements in relation to their technical and tactical skills [40,41]. In fact, basketball games are characterized by short bouts of high-intensity activity interspersed by low- to moderate-intensity ones, which involve repeated defense-to-offence transitions [42,43,44]. The external load of basketball activities differs in terms of movement patterns (i.e., fast accelerations, abrupt decelerations, running, jumping, shuffling, and changing directions), distance, frequency, and duration, eliciting an internal load relying on both aerobic and anaerobic energy production systems. Unfortunately, only a few studies have focused specifically on masters basketball players (<65 years), mainly during amateur games [13,37]. When compared to co-aged soccer players, masters basketball players showed lower aerobic capacity and 10 m sprint performances, and a higher jumping capability, thus confirming the sport-specific effects [37]. Nonetheless, after 50-min friendly basketball games, the senior players significantly increased their jump and their in-phase interlimb coordination performances and maintained their sprint and anti-phase interlimb coordination capabilities, which confirmed that the practice of sports mitigates the age-related regression of functional fitness [13].

Despite the fact that internal and external game loads in youth and elite basketball have been extensively investigated [44], little information is available on masters players [37]. An investigation of amateur basketball players aged 55 to 62 years undergoing a training regimen of 1.5 h per week^–1^ with a friendly game reported a generally high game intensity and significant occurrence of high heart rate (HR) frequencies, with players showing good anaerobic characteristics and a moderate aerobic capacity [37]. Actually, basketball is included among physical activities that must be employed cautiously when it comes to high-risk, low-fit, and/or symptomatic cardiorespiratory older individuals [2,45]. Therefore, there is a need for multidimensional knowledge encompassing the internal and external game loads of masters basketball games to structure a tailored training program for older basketball players engaged in championships organized at national levels [46]. 

Therefore, to contribute to the growing literature on senior sports, the present study aimed to assess the internal and external load of masters basketball athletes during official games of a national championship.

## 2. Materials and Methods

### 2.1. Participants

Thirteen male basketball masters athletes (age: 66.6 ± 2.1 years; range: 65–79 years; body mass: 89.9 ± 8.7 kg; stature: 183.7 ± 4.6 cm) were recruited for this study. The participants were highly experienced basketball players, with 51.5 ± 3.8 years competing at professional (15%) and amateur (85%) levels, and with at least 2 years of experience in official masters basketball championships. The participants were recruited from three masters basketball teams competing in the championship organized by the Lithuanian Basketball Veteran League [Lietuvos Krepsinio Veteranu Lyga (LKVL)] for the over-65 category. The participants were informed about the study aims and procedures before providing personal written informed consent. Ethical approval was received by the Lithuanian Sports University Review Board (code: SMTEK-124) in accordance with the ethical standards of the Helsinki Declaration.

### 2.2. Design

This observational study was conducted during the 2019–2020 LKLV season. The LKLV encompasses several older-age categories, with the over-65 category including four teams competing in the regular season and in the final phase. The games were played according to the FIBA rules, which encompass four 10 min quarters interspersed with a 15 min half time break and 2 min between the other quarters. Furthermore, typical basketball stoppage and live times with a shot-clock set at 24 s for each ball possession were adopted. The regular phase of the championship was organized in three 2-day meetings held between December 2019 and February 2020, during which all the teams played 2-day consecutive games, and the final phase was organized in March 2020. Overall, the performance profile of the 13 participants was assessed during their first game of the first 2-day meeting of the LKVL regular phase to avoid any possible fatigue effect due to playing two basketball games in consecutive days [47]. Considering that during the games the players could be on and off the court with different plying times, the total duration (i.e., from the starting tip-off to the end of the game, including all stoppages) and the actual game time of the individual players (i.e., the time the players spent on the court, excluding between-quarter periods, time-outs and bench time with all other in-game stoppages for out-of-bounds and free-throws included) were recorded for each game [47].

### 2.3. External Load

Prior to the observed game, the players were individually equipped with inertial measurement units (IMU) (Catapult ClearSky T6, Catapult Innovations, Melbourne, VIC, Australia). IMU devices were placed in manufacturer-provided vests for secure attachment onto each player between the scapulae and worn under competitive sportswear. After collection, the data were exported to Catapult proprietary software (Catapult Openfield, v1.18, Catapults Innovations, Melbourne, VIC, Australia) for further analysis. External workload measures included absolute PlayerLoad (PL), which was calculated as the instantaneous change rate in accelerations using the triaxial accelerometer component of the microsensors sampling at 100 Hz with the following formula: PL (AU) = [√(Ac1_n_ − Ac1_n−1_)^2^ + (Ac2_n_ − Ac2_n−1_)^2^ − (Ac3_n_ − Ac3_n−1_)^2^] ∗ 0.01, where Ac1, Ac2, and Ac3 are the orthogonal components measured from the triaxial accelerometer and 0.01 is the scaling factor [48]. Further IMUs included accelerations (ACC), changes of direction (COD), and jumps (JUMPS). The ACC and COD were categorized as high (>3.5 m · s^−2^), medium (2.5–3.5 m · s^−2^) and low (1.5–2.5 m · s^−2^) on the basis of their intensities [49]. Jumps were categorized as low (<20 cm) and high (>20 cm). Each IMU variable is used extensively in basketball [49,50,51,52,53]. Each IMU variable was also reported relative to the actual game time (PL/min; ACC/min; COD/min; JUMP/min).

### 2.4. Internal Load

Objective internal load was assessed as percentages of HR_peak_ (%HR_peak_) measured via HR chest-worn monitors (H10, Polar Electro, Kempele, Finland). For each player, the HR_peak_ was considered as the highest HR value recorded during the assessed game. In fact, HR_peak_ was previously used for basketball players when assessments of HR_max_ by means of court-based or lab-based basketball specific tests were not feasible [37,54]. During each game the HR monitor was synchronized with the IMU device, and the data were downloaded and stored at the end of the game (Catapult Openfield, v1.18, Catapults Innovations, Melbourne, VIC, Australia). Due to technical problems, the HR of 5 of the players was not properly recorded; therefore, the HR values for 8 players only were included in the statistical analysis. 

Subjective internal load was measured using the session rating of perceived exertion (sRPE) method, which has been extensively used in basketball research [47,55,56,57,58,59,60,61,62,63,64]. Each player was individually required to provide a global intensity score using the Borg CR-10 modified scale [65] approximately 30 min after each game by answering the question: “How intense was the game?” [55]. The sRPE load was then calculated by multiplying the sRPE score by the game duration (i.e., total and actual time) in minutes [47]. The sRPE scores were collected using the paper-and-pencil method since older athletes might have had difficulties using cloud-based online survey software, which has been showed to possess a stronger association with heart rate–based training methods compared to the paper- and-pencil method [64].

### 2.5. Statistical Analysis

Mean ± SD were used as descriptive statistics for each dependent variable. Moreover, frequency of occurrence was calculated as counts (calculated summing of the events of each player), and percentages were assessed for the external load measures referred to JUMP, ACC, and COD. Successively, a chi-square goodness-of-fit test was adopted to assess differences in distribution within the JUMP, ACC, and COD classes of intensities (i.e., low, medium, and high). The effect size was assessed using the φ value and interpreted following Cohen’s benchmarks considering 0.1, 0.3, and 0.5 as small, medium, and large effect size, respectively [66]. All the data were calculated using SPSS software (v. 26), and the level of significance was set at *p* < 0.05.

## 3. Results

The results revealed that PL = 269.9 ± 83.3 AU and PL/min = 6.54 ± 1.29 AU∙min^−1^. All other external load measures referring to load volume and intensities are displayed in Table 1 and Table 2, respectively. A significant difference (*p* < 0.001) in the distribution of the intensity classes was found for JUMP, ACC, and COD with large effect sizes and the highest frequency of occurrence for the low-intensity classes and the lowest frequency of occurrence for the high-intensity ones (Table 1).

The analysis of internal load showed HR_peak_ = 81.7 ± 8.1% and sRPE-load of 148.9 ± 69.7 AU and 259.7 ± 71.8 AU for actual (41.6 ± 11.9 min) and total (74.8 ± 6.4 min) game duration, respectively.

## 4. Discussion

The aim of this study was to assess the internal and external loads of official masters basketball games played during a national competition. The main results indicated that masters basketball games are characterized by low external load intensity (i.e., PL/min) compared to younger basketball populations. Furthermore, ACC, COD, and JUMP presented the highest frequency of occurrence for the low-intensity classes and the lowest occurrence for the high-intensity classes. Despite the fact that the players reported a low perceived exertion (i.e., sRPE), a high objective internal load response (i.e., %HR_peak_) emerged. Overall, these data are novel in the literature for senior basketball athletes and can provide senior-basketball coaches new insight for designing sound training sessions.

To meet situational tactical scenarios within a basketball game, players are required to have cognitive and decision-making skills as well as the performance of sport-specific movements encompassing variable velocities, accelerations, decelerations, stopping, and sudden changes of direction [67]. The unpredictable and intermittent nature of basketball games due to complex interactions among game-related determinants at physical (e.g., age, technical capabilities, fatigue status of the players), technical (e.g., playing positions), and tactical (e.g., individual and team) levels urge specific investigations to identify relevant aspects to construct sound training plans for senior basketball players [37,44,46]. Information on older players is limited, obtained during amateur friendly games, and mainly based on the evaluation of the internal load (HR and RPE), motion analysis performed through video recordings (e.g., running, walking, inactivity, positioning, and jumping) due to lack of proper wearable equipment, and post-game acute effects on jump, sprint, strength, and coordination performances [14,37]. Current technological advancements in the field of wearable sensors enable the quantification of internal and external load during actual competitions. To the best of our knowledge, this is the first investigation assessing the occurrence of ACCs, CODs, and JUMPs during official senior-basketball games by using wearable electronic equipment. 

The present analysis of the external load measured via IMUs during official games of a Basketball Veteran League competition showed that the players mainly performed at low intensity. Various investigations assessed the external game load via IMUs across several basketball populations [60,61,62,68,69]. However, the different experimental approaches, measurement devices, and data reduction in relation to the whole and/or actual game time (i.e., full match duration including stoppages or only the time the players were on-court) do not allow sound comparisons between studies. 

In considering the only two studies based on the same experimental approach and data reduction [47,49], the present older players showed lower PL/min values (6.54 ± 1.29 AU∙min^−1^) with respect to professional (11.1 ± 2.0 AU∙min^−1^) and semi-professional (11.6 ± 1.5 AU/min) counterparts, and lower frequency of occurrence of JUMP (total JUMP: 0.28 ± 0.13 n∙min^−1^) and ACC (total ACC: 0.60 ± 0.21 n∙min^−1^) with respect to professional players (total JUMP: 1.11 ± 0.53 n∙min^−1^; total: ACC: 2.19 ± 0.84 n∙min^−1^). Although senior athletes present better physical functioning with respect to their co-aged sedentary counterparts, a decrease in high-intensity actions requiring good strength and power production is expected because of the decline over time due to biological aging [17,37]. In fact, Tessitore et al. [37] found a 62% lower jumping ability in senior basketball players (55 ± 5 years) compared to elite players. Overall, the present investigation presents a considerable decline in the frequency of occurrence of the external load parameters imposed by basketball games in male, older adult basketball players and can provide benchmarks for coaches and practitioners to design appropriate training sessions and drills aiming to replicate the game intensities. The execution of ACCs, CODs, and JUMPs has been suggested to benefit bone health in older adults [70], particularly during team sport activities [71]. Interestingly, the present findings revealed that during a basketball game older players most frequently perform CODs, JUMPs, and ACCs at low intensity (large effect size). Conversely, it has to be considered that the total external load results are particularly high for older adults. In fact, the basketball players performed a substantially higher number of low-intensity CODs (n = 1759) and ACCs (n = 205) compared to a session of walking football (~100 COD and ~45 ACCs), which is a team sport activity specifically created for adults and older adults [71]. It can be speculated that ACCs, CODs, and JUMPS performed during senior basketball games produce a large ground/joint reaction force in older athletes, which might be beneficial for their bone density [70] but might expose them to a high risk of injury [72]. Conversely, a recent meta-analysis on jumping exercise protocols in healthy older individuals claimed a relative safety and low injury incidence of power-based exercises especially under supervision of qualified coaches [72]. Therefore, the present study could provide relevant information for coaches of older basketball teams to structure sound training programs preserving the physical functioning of the players while reducing the risk of injuries during games. Furthermore, future investigations assessing the differences in bone density and injury occurrence in older amateur and competitive basketball players are strongly suggested. Similar to recreational soccer, future studies should also investigate whether basketball activities could be beneficial for lower limb bones in untrained older individuals [73].

The external load intensities recorded during the investigated senior basketball games induced a high physiological demand (>80%HR_peak_), similarly to previous investigations showing that friendly senior basketball games elicited a high cardiovascular load [13,37]. In general, exercise involving the aerobic/anaerobic systems helps to reduce health risks, being linked with healthy blood vessels and cardiovascular profiles [74,75]. However, the HR of around 80% of HR_peak_ that emerged from the present analysis of the internal loads could increase apprehensions for the basketball players’ health risks in competitions [76]. Therefore, to avoid potential cardiovascular risks for older basketball players, they should run frequent health checkups, particularly for players reporting risk factors for ischemic heart disease or sudden death. Despite a high physiological demand, senior basketball players showed low sRPE values. In the present study, the players perceived the intensity of their official game as moderate, with lower RPE values with respect to those reported by recreational older basketball players at the end of a friendly game [13,37]. Overall, recreational basketball activities have been suggested to produce a low perceptual demand in recreational adult participants [77]. It could be speculated that the enjoyment associated with playing official basketball competitions might moderate the subjective perception of the actual demand of the game. This is a crucial point for older basketball players, since the low perceived exertion despite a high physiological response can have the potential to motivate players to become engaged in physical activity and in turn increase their adherence to long-term basketball training programs. Indeed, participation in sports could enhance the wellbeing of older individuals through enjoyment and social connection, and moving is crucial to adherence [78].

Although this study provides novel findings and provides an interesting snapshot documenting the external and internal loads of a senior basketball game, some limitations should be acknowledged. Firstly, we investigated only one game, and the assessment of multiple games could provide more insight into the performance profile of senior-basketball competitions; this is particularly true of competitions played in close succession, as they represent a typical tournament scenario. Moreover, the technical and tactical demands of senior basketball games were not evaluated, but this might imply different loads across games. Therefore, future studies should investigate a larger number of basketball games, including those played in tournament style, along with an analysis of the technical and tactical demands of the game.

## 5. Conclusions

A senior basketball game is characterized by a relatively low external load intensity compared to professional or semi-professional basketball players but with a high number of total occurrences for senior athletes. Additionally, a senior basketball game elicited a high physiological demand measured with %HR_peak_, which suggests the necessity of frequent health checks for senior athletes involved in basketball activities. Finally, the high physiological demand does not match the relatively low perceived demand elicited by a senior basketball game, which could increase the players’ adherence to compete in basketball tournaments.

## Figures and Tables

**Table 1 jfmk-07-00111-t001:** Descriptive statistics and chi-square analysis for external load measures referred to game volume.

External Load	Intensity	Mean ± SD (n/Game)	Frequency of Occurrence (n)	Frequency of Occurrence (%)	χ^2^	*p*	φ
JUMP	High	9.8 ± 6.7	128	79	54.4	<0.001	0.58 (Large)
	Low	2.6 ± 2.8	34	21			
	Total	12.5 ± 7.0	162	100			
ACC	High	3.5 ± 2.3	46	13.7	123.5	<0.001	0.61 (Large)
	Medium	6.5 ± 4.0	84	25.1			
	Low	15.8 ± 8.6	205	61.2			
	Total	25.8 ± 13.1	335	100			
COD	High	5.6 ± 5.2	73	3.5	2474.1	<0.001	1.09 (Large)
	Medium	19.2 ± 12.1	250	12			
	Low	135.3 ± 63.5	1759	84.5			
	Total	160.2 ± 79.1	2082	100			

Note: SD = standard deviation; χ^2^ = chi-square value; *p* = probability value; φ = phi-value (effect size); JUMP-high = jumps > 20 cm; JUMP-low = jumps < 20 cm; JUMP-total = total number of jumps; ACC-high = accelerations at >3.5 m · s^−2^; ACC-medium = accelerations at 2.5–3.5 m · s^−2^; ACC-low = accelerations at 1.5–2.5 m · s^−2^; ACC-total = total number of accelerations; COD-high = changes of direction at >3.5 m · s^−2^; COD-medium = changes of direction at 2.5–3.5 m · s^−2^; COD-low = changes of direction at 1.5–2.5 m · s^−2^; COD-total = total number of changes of direction.

**Table 2 jfmk-07-00111-t002:** Mean ± SD of external load measures according to their intensity.

External Load	Intensity	Mean ± SD
JUMP (n∙min^−1^)	High	0.22 ± 0.14
	Low	0.06 ± 0.06
	Total	0.28 ± 0.13
ACC (n∙min^−1^)	High	0.08 ± 0.04
	Medium	0.02 ± 0.01
	Low	2.41 ± 1.24
	Total	0.60 ± 0.21
COD (n∙min^−1^)	High	0.12 ± 0.09
	Medium	0.07 ± 0.03
	Low	20.70 ± 9.23
	Total	31.40 ± 34.48

Note: SD = standard deviation; JUMP-high/min = jumps >20 cm per minute; JUMP-low/min = jumps <20 cm per minute; JUMP-total/min = total number of jumps per minute; ACC-high/min = accelerations at >3.5 m · s^−2^ per minute; ACC-medium/min = accelerations at 2.5–3.5 m · s^−2^ per minute; ACC-low/min = accelerations at 1.5–2.5 m · s^−2^ per minute; ACC-total/min = total number of accelerations per minute; COD-high/min = changes of direction at >3.5 m · s^−2^ per minute; COD-medium/min = changes of direction at 2.5–3.5 m · s^−2^ per minute; COD-low/min = changes of direction at 1.5–2.5 m · s^−2^ per minute; COD-total/min = total number of changes of direction per minute.

## Data Availability

The data presented in this study are available on request from the corresponding author.

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
