# Peer review of "Investigating External and Internal Loads in Male Older Adult Basketball Players during Official Games"

_jfmk, 2022, doi:10.3390/jfmk7040111_

Round 1

Reviewer 1 Report

The paper entitled Investigating external and internal loads in male older adult basketball players during official games by
Daniele Conte , Federico Palumbo , Flavia Guidotti * , Kestutis Matulaitis , Laura Capranica , Antonio Tessitore
is interesting and in general well written however should be proofreaded by native speaker

The paper is well written, and I am not suprised the issue since Litjhuania is a source of the most talented baskatball players all over the world and this discipline is very common and popular in this country so the Authors have abslotutely high experience in this issue.
 Title is correct
Materials and methods are clear and have no mistakes

Unfortunately, the weakness of this well-constructed study is that the results and conclusion are rather obvious. I really don't know if the relatively low novelty of this research justifies publishing in journals about IF=6plus

Author Response

We do that the Reviewer 1 for his/her precious comments. We provided more information regarding the rationale of our study in the attached file.

Author Response

We thank the Reviewer 2 for his/her precious comments, which we have addressed point-by-point in the attached file.
